# Point-of-Care Testing for Sensitive Detection of the African Swine Fever Virus Genome

**DOI:** 10.3390/v14122827

**Published:** 2022-12-19

**Authors:** Ahmed Elnagar, Sandra Blome, Martin Beer, Bernd Hoffmann

**Affiliations:** Institute of Diagnostic Virology, Friedrich-Loeffler-Institut, 17493 Greifswald-Insel Riems, Germany

**Keywords:** African swine fever virus, DNA isolation, portable real-time PCR, point-of-care (POC)

## Abstract

African swine fever (ASF) is a contagious viral hemorrhagic disease that affects domestic pigs and wild boar. The disease is notifiable to the World Organization of Animal Health (WOAH), and causes significant deaths and economic losses. There is currently no fully licensed vaccine available. As a result, early identification of the causative agent, ASF virus (ASFV), is crucial for the implementation of control measures. PCR and real-time PCR are the WOAH-recommended standard methods for the direct detection of ASFV. However, under special field conditions or in simple or remote field laboratories, there may be no sophisticated equipment or even stable electricity available. Under these circumstances, point-of-care systems can be put in place. Along these lines, a previously published, rapid, reliable, and electricity-free extraction method (Triple*E*) was used to isolate viral nucleic acid from diagnostic specimens. With this tool, nucleic acid extraction from up to eight diagnostic samples can be realized in one run in less than 10 min. In addition, the possibility of completely omitting viral DNA extraction was analyzed with so-called direct real-time PCR protocols using ASFV original samples diluted to 1:40 in RNase-free water. Furthermore, three real-time PCR cyclers, developed for use under field conditions (IndiField, Liberty16 and UF-300 Genechecker^TM^), were comparatively applied for the sensitive high-speed detection of ASFV genomes, with overall PCR run times between 20 and 54 min. Depending on the viral DNA extraction/releasing method used and the point-of-care cycler applied, a total time for detection of 30 to 60 min for up to eight samples was feasible. As expected, the limitations in analytical sensitivity were positively correlated to the analysis time. These limitations are acceptable for ASFV diagnostics due to the expected high ASFV genome loads in diseased animals or carcasses.

## 1. Introduction

African swine fever virus (ASFV) is the only member of the *Asfarviridae* family and the genus *Asfivirus*. It is a complex double-stranded DNA virus with a size of 170–190 kbp, and around 151 to 167 open reading frames [1]. It is the causative agent of African swine fever (ASF), which only affects *Suidae*. ASF is a fatal disease that can cause death in up to 100% of infected domestic pigs and wild boar of the species *Sus Scrofa* [2]. It has generated enormous economic losses in the pig industry, especially since 2007 [3]. In Africa, argasid ticks of the genus *Ornithodoros* can spread the virus [4], although outside of Africa, transmission via direct contact with infected animals or carcasses is the most relevant way of transmission.

ASF outbreaks in Asia and Europe have killed millions of pigs, and the disease has recently been spreading throughout different countries in Europe (Bulgaria, Czech Republic, Hungary, Moldova and Romani, Belgium, Poland, and since 2020, Germany) [5]. There is currently no effective vaccination or therapy for ASF; hence, the early and swift detection of this virus is critical for any control measures.

Real-time polymerase chain reaction (PCR) is one of the fastest and most sensitive laboratory procedures for detecting pathogen nucleic acid material in clinical samples. Therefore, conventional and real-time PCR are considered to be reliable methods for ASFV detection [6,7], and are recommended by the WOAH. In addition, PCR has been shown to be an excellent and rapid technique that can be used as a routine diagnostic tool for ASFV in surveillance, control, and eradication programs [6,7,8,9,10,11].

Point-of-care testing (POCT) has been developed to provide more efficient disease control and a reliable diagnostic tool under special field conditions, without the need to send samples to specialized or central diagnostic laboratories. POCT can be particularly useful for disease diagnosis in remote areas, where infrastructure and laboratory capacity are limited [9].

Rapid antigen detection tests, such as lateral flow devices (LFDs), are easy to use under field conditions, but their current diagnostic performance has not yet been highly standardized, and especially their diagnostic sensitivity is reduced [12,13]. Various approaches have been described for POCT, but the most reliable solution still seems to be genome-based systems, especially in combination with simple extraction procedures. Loop-mediated isothermal amplification assays have the potential for field diagnosis of ASF, but concerns with either their diagnostic performance for clinical samples or their risk of contamination may have limited their wider application [14,15]. The problem of fast, efficient, and electroless nucleic acid extraction in the field must be clarified for all genome-based detection methods. In this context, Korthase et al. (2022) established a rapid and electricity-free extraction method applicable for all POCT that detects pathogen-specific RNA/DNA [16]. A recombinase polymerase amplification (RPA)-based method was reported as a simple, cost-effective, and fast diagnostic tool for rapid and specific detection of ASFV, as described by the study of Wang et al. (2017) [17]. Furthermore, Daigle et al. (2020) described the successful transfer of a highly sensitive and specific laboratory-validated real-time PCR assay to a portable pen-side thermocycler, which can be operated in the field for rapid detection of ASFV following quick manual nucleic acid extraction [18]. Other studies have developed highly sensitive and specific real-time PCR assays that have been validated and used in diagnostic laboratories around the world for the detection of genetic material in clinical samples [19,20,21,22]. Briefly, it has been demonstrated that the most reliable solution still seems to be PCR-based methods, especially in combination with simple extraction procedures.

All of these molecular tests could help in epidemiological investigations for diagnosing the disease in remote areas with sparse infrastructure and limited laboratory capacity. In addition, screening of wild boar carcasses directly at the site of discovery could save time and resources [13]. The transport of clinical samples to diagnostic laboratories in remote areas can take a longer time, prolonging the process of diagnosis and delaying the results needed for a rapid response.

The objective of our study was to evaluate molecular diagnostic tools for the so-called point-of-care (POC) concept. For this purpose, suitable methods for rapid and simple ASFV DNA extraction and release were tested, and different real-time POC PCR cyclers were analyzed comparatively. The recently described electricity-free hand extraction method (*Easy Express Extraction*, Triple*E* system) [16], which was able to isolate up to eight samples in less than 10 min, was validated in comparison to an automated routine extraction system that was also based on magnetic bead technology (IndiMag 48).

In addition, direct qPCR—based on the 1:40 dilution of ASFV-positive clinical specimens in water—was also performed. For specific genome detection, three portable real-time PCR thermal cyclers (IndiField from Indical Bioscience, Liberty16 from Ubiquitome, and UF-300 Genechecker^TM^ from Genesystem) were compared with a standard real-time PCR cycler for the laboratory (CFX96 from Bio-Rad). The aim was to analyze the basic suitability of these POC cyclers for the sensitive detection of ASFV genomes under strongly reduced time conditions.

## 2. Materials and Methods

### 2.1. Sample Collection

The panel consisted of 34 samples from domestic pigs and wild boar that had been collected in four different animal experiments, with ASFV strains of different genotypes. The animals were housed in groups at a high-containment facility in the Friedrich-Loeffler-Institut (FLI) (L3+). The animals were fed a commercial pig food with corn and a hay cob supplement, and had access to water ad libitum. The animal trials were approved by a competent authority (Landesamt für Landwirtschaft, Lebensmittelsicherheit und Fischerei (LALLF) Mecklenburg-Vorpommern, Rostock, Germany) under reference number 7221.3-2.011/19. For the analyses, sample matrices from 22 animals that were infected with several ASF virus strains were used (10 EDTA blood samples (Estonia 2014), one EDTA blood sample (CHZT 90/1), three EDTA blood samples (Belgium 2018), eight EDTA blood samples (SUM 14/11), two lung tissue samples, four spleen tissue samples, four liver tissue samples (all organ samples from a trial with Estonia in 2014), and eight bone marrow tissue field samples). The samples reflected different routine matrices and had been collected at different time points post-infection. Furthermore, eight bone marrow samples were collected from wild boar carcasses from actual outbreaks in Germany. The latter samples were delivered from the state laboratory by the local authority of the outbreak region. Overall, 42 specimens were used in this study (Appendix A).

### 2.2. DNA Extraction/Releasing Methods

The IndiMag 48 platform and an IndiMag^®^ Pathogen Kit (both INDICAL BIOSCIENCE, Leipzig, Germany) were used as the standard automated extraction method for comparative purposes, as described in the study of Elnagar et al. (2021) [23].

Next, compared to the standard automated extraction system, we validated a recently described manual extraction method that does not need any electricity or centrifugation steps, which can therefore easily be performed under field conditions. This Easy Express Extraction (Triple*E*) system represents a fast and affordable magnetic bead-based extraction method that is also based on the IndiMag^®^ Pathogen Kit (INDICAL BIOSCIENCE, Leipzig, Germany). We used this method, as described by Korthase et al. (2022) [16]. Compared to both extraction methods, we conducted direct qPCR amplification of the original materials by its dilution to 1:40 in RNase-free water, mixing it well by pipetting up and down, and subsequently used it directly without further treatment as a PCR template.

EDTA blood was carefully mixed several times before viral DNA extraction/releasing started. The tissue samples were homogenized by grinding approximately 0.5 g of organ tissue with a 5 mm steel ball within 1 mL of cell culture medium in 2 mL bolted tubes that were shaken up and down more than 30 times. The liquid supernatant of the homogenate was used for further processing.

### 2.3. Real-Time PCR Detection Systems

#### 2.3.1. CFX 96 Standard System

The ASFV qPCR assay described by Haines et al. [11] was modified by the integration of a lab-specific internal control system [24]. For the amplification, a PerfeCTa^®^ qPCR ToughMix^®^ Kit from Quanta BioSciences (Gaithersburg, MD, USA) was applied. Briefly, a FAM-labeled ASFV primer–probe mixture consisting of 800 nM primer ASFVp72IVI-F (5′-GAT GAT GAT TAC CTT YGC TTT GAA-3′), 800 nM primer ASFV-p72IVI-R (5′-TCT CTT GCT CTR GAT ACR TTA ATA TGA-3′), and 200 nM probe ASFV-p72IVI-FAM (5′-FAM-CCA CGG GAG GAA TAC CAA CCC AGT G-BHQ1-3′) in 0.1 × TE buffer (pH 8.0) was realized. For control of the extraction and qPCR, a heterologous control system, published by Hoffmann et al. (2006) [24], was integrated. Here, a HEX-labeled primer–probe mixture consisting of 200 nM primer EGFP1-F (5′-GAC CAC TAC CAG CAG AAC AC-3′), 200 nM primer EGFP2-R (5′-GAA CTC CAG CAG GAC CAT G-3′), and 200 nM EGFP-probe 1 (5′-HEX-AGC ACC CAG TCC GCC CTG AGC A-BHQ1-3′) in 0.1 × TE buffer (pH 8.0) was prepared. Then, 12.5 μL of the total reaction mix was established with 1.75 μL of RNase-free water, 6.25 μL of 2 × PerfeCTa qPCR ToughMix, 1.0 μL of the ASFV primer–probe mixture (ASFV-P72-IVI-Mix-FAM), 1.0 μL of the internal control primer–probe mixture (EGFP-Mix1-HEX), and 2.5 μL of the DNA template. The following thermoprofile was used for the amplification: 3 min at 95 °C, 45 cycles at 95 °C for 15 s, 60 °C for 20 s, and 72 °C for 20 s. The fluorescence data in the FAM and HEX channels were collected during the annealing step, and the total run time on the CFX96 real-time detection system (Bio-Rad, Hercules, CA, USA) could be ascertained as 1 h and 16 min. For data analyses, Bio-Rad Maestro software (Version: 4.1.2433.1219) was used.

#### 2.3.2. IndiField PCR System

The IndiField PCR system (INDICAL BIOSCIENCE, Leipzig, Germany) is an ultra-portable thermocycler that weighs around 1.4 kg, and has a rechargeable battery with a lifespan of approximately 8 h. It is fully controlled by a smartphone, holds up to nine samples, and has the ability to detect 27 analytes in parallel (three fluorescence channels per well). An IndiField ASFV PCR Kit (INDICAL BIOSCIENCE, Leipzig, Germany) was used, which was prepared as ready-to-use lyophilized reagents in the individual PCR tubes of the IndiField thermocycler. The reaction mix was prepared by adding 20 µL of the DNA template directly to the lyophilized master mix, including the ASFV target assay (FAM channel) and the internal control assay (ROX channel). The PCR data can be uploaded to a cloud-based storage and analysis system. A PCR thermoprofile of 1 min at 95 °C, followed by 45 cycles at 95 °C for 1 s and 60 °C for 20 s, was introduced by scanning the specific QR code on the package of the lyophilized IndiField ASFV PCR Kit [23]. The total run time for this system on the IndiField thermocycler was 56 min.

#### 2.3.3. Liberty16 PCR System

The Liberty16 PCR system (Ubiquitome, New Zealand) is an easy and fast thermocycler (FAM channel only) with an outside dimension of 3.2 kg; it is provided with an internal rechargeable lithium-ion battery. Here, a Biozym Blue Probe qPCR Mix Separate ROX (Biozym, Hessisch Oldedorf, Germany) was used for amplification. A total reaction of 12.5 µL consisted of 2.75 µL of water, 6.25 µL of 2 × Blue Probe qPCR ToughMix, 1.0 μL of the ASFV primer–probe mixture (ASFV-P72-IVI-Mix-FAM), and 2.5 µL of the DNA template. The PCR data do not require a laptop to be run; however, the Ubiquitome iPhone app does need to be downloaded from the App Store. This app allows for setting up the run, viewing the run-in progress, calling Cq dynamic graphing of the annotated real-time PCR amplification curves, and uploading data to share in the cloud. The PCR run was performed via a Bluetooth connection with a thermal profile of 1 min at 95 °C, followed by 40 cycles at 95 °C for 3 s and 60 °C for 3 s. The total run time for this Liberty16 PCR system was 37 min.

#### 2.3.4. Genechecker UF-300 PCR System

The UF-300 Genechecker^TM^ dual channel real-time PCR system (Genesystem Co., Daejeon, Korea) is a compact and intuitive platform (3.3 kg) for point-of-care molecular diagnostics with available dual detection channels (FAM/ROX). For application in the field, the thermocycler can be operated via a vehicle cigarette lighter. The system has a touch panel interface (8 inches) so that users can intuitively set the parameters and instantly run tests. The screen consists of four simple menus, and test protocols can be pre-programmed for immediate startup. The system can finish a PCR run within 20 min. The high ramping rates for heating and cooling are based on a special microfluid PCR chip associated with a compact and sophisticated hardware mechanism. The microfluid chip has a capacity of 10 samples per PCR run. A Biozym Blue Probe qPCR Mix Separate ROX (Biozym, Hessisch Oldedorf, Germany) in a total reaction volume of 10 µL was also applied here for amplification. Finally, 5 µL of 2× Blue Probe qPCR ToughMix, 2.0 μL of the ASFV primer–probe mixture (ASFV-P72-IVI-Mix-FAM), and 3 µL of the DNA template were mixed for one well. A PCR thermal profile of 1 min at 95 °C, followed by 40 cycles at 95 °C for 3 s and 60 °C for 3 s, was performed. The run time of the UF-300 was 19 min.

## 3. Results

A comparison of the three different nucleic acid extraction/releasing methods and four qPCR systems was performed to acquire a broad applicability range for ASFV DNA isolation and genome amplification in the field (Figure 1, Figure 2 and Figure 3). All of the tested samples were first extracted with the IndiMag 48 system and amplified with the Bio-Rad CFX96 standard system using the in-house Haines qPCR (Haines assay), in order to generate qualitative and quantitative reference data (Appendix A). For the POCT, nucleic acid extraction/releasing, the Triple*E* system as well as direct qPCR amplification (samples diluted 1:40 with distilled water) were comparatively tested. In addition, the extracted/released ASFV DNA was tested by applying three different POCT thermocyclers (IndiField, Liberty16, and UF-300).

### 3.1. Qualitative Data Analysis

In terms of the Ct values based on the standard PCR amplification on the Bio-Rad CFX 96, we divided the dataset of 42 samples overall into four groups, in order to determine and evaluate the efficacy and sensitivity of each portable PCR thermocycler based on the different extraction methods.

Group I comprised samples with Ct values between 15 and <20 (11 samples); Group II (Ct 20–<25) included 11 samples; Group III (Ct 25–<30) comprised 13 samples; and Group IV included 7 samples with Ct values higher than 30 (Table 1 and Appendix A). Qualitative data evaluation based on the different extraction and qPCR methods showed very clearly that positive ASFV detection is dependent on the viral genome load in the different samples. All of the high-load Group I and II samples with Ct values between 15 and 25 could be successfully detected, regardless of the extraction method or the qPCR cycler used. The Group III samples (Ct 25–30) could always be detected when extracted with the IndiMag 48 or Triple*E* extraction system. Even when using direct qPCR, the samples of Group III could be successfully amplified in the vast majority of cases (all samples were detected positive). Only the combination of direct PCR and the Liberty16 cycler yielded a negative result for 4 of the 13 samples of the moderately loaded Group III. A similar result was obtained with the weak positive samples of Group IV (Ct > 30). Here, most of the samples could be successfully detected after extraction with the IndiMag 48 or Triple*E* method (24 and 19 of the 28 samples, respectively). In contrast, ASFV detection of these samples after direct qPCR was positive in only one of the 28 tests (Table 1). An overview about the estimated time of each DNA extraction and PCR amplification run was summarized in (Table 2).

### 3.2. Quantitative Data Analysis

Based on the standard automated IndiMag 48 extraction system, all data obtained by the four qPCR systems showed comparable results in terms of Ct values, with a slightly higher sensitivity for the IndiField PCR system using the ASFV IndiField PCR Kit. This trend was confirmed with samples extracted with both POC extraction/releasing systems. Using the Triple*E* hand extraction system and the direct qPCR amplification, the quantitative results were similar among all tested samples. The ASFV IndiField PCR also showed the lowest Ct values and highest sensitivity.

Figure 1 shows the comparative mean Ct values for the different sample matrices as a function of the extraction method. The raw data for the analyses were compiled, and are presented in Appendix A. A one-way ANOVA was performed to test the significance of the different PCR systems being compared, with a resulting *p*-value of >0.99 for all samples taken, which is not statistically significant.

From the analyses, it can be seen that regardless of the ASFV strains or matrices tested, there was very good agreement between Ct values. The direct qPCR amplification presented less sensitivity than the data obtained using extracted DNA for qPCR.

Nevertheless, the highly simplified releasing procedure without any need for extraction was also able to detect the pathogen with acceptable Ct values, especially in samples with high viral loads.

The two different nucleic acid extraction methods delivered very similar results, demonstrating that the electricity-free hand extraction (Triple*E*) could be a very suitable component of the molecular POC testing procedure. Furthermore, the data showed that the two magnetic bead-based extraction methods are quite comparable; however, the automated IndiMag 48 platform had higher sensitivities. In addition, direct qPCR amplification presented a statistically significantly lower sensitivity compared to the IndiMag 48 extraction. Interestingly, the samples were classified correctly with all four different real-time PCR systems (Figure 2).

## 4. Discussion

African swine fever is one of the most serious viral infections of domestic pigs and wild boar, and has a tremendous impact on animal health and the pig industry. Due to the lack of vaccination or treatment options, early detection is of utmost importance to recognize outbreaks and apply control measures as soon as possible [25]. Domestic pigs and Eurasian wild boar show severe clinical manifestations after ASFV infection [26]. Since most clinical signs are very unspecific, laboratory testing is required to corroborate any clinical suspicions [19]. Here, two different DNA extraction systems were evaluated and compared to the performance of extraction-free direct qPCR amplification as a diagnostic tool in the field. Direct qPCR was performed as an alternative to DNA isolation methods using the output of the diluted original samples as a template for PCR amplification. This approach was validated to suit the field application against the various standard DNA extraction methods and PCR systems. It is clear that direct qPCR without prior nucleic acid extraction has limitations, as inhibitors present in the sample can influence the performance of the qPCR. Newly developed master mixes, such as the Biozym Blue Probe qPCR Mix, show improved tolerance to inhibitory substances. Nevertheless, genome amplification from blood is a particular challenge, as hemoglobin is considered to be a potent PCR inhibitor. It should also be noted that the inhibitor tolerance of DNA-dependent DNA polymerases is higher than that of RNA-dependent reverse transcriptases. Thus, the meaningful and successful use of direct qPCR is particularly dependent on the sample matrix, but also on whether (viral) RNA is to be detected in addition to DNA. The limitations of direct qPCR require qualified technical staff, which in turn can be a significant obstacle in the field [27].

These difficulties may be overcome by employing an on-site hand nucleic acid extraction tool. Manual processing of the magnetic beads bypasses this technological barrier, allowing the extraction process to include numerous additional washing steps. The performance of the so-called Triple*E* system has been shown previously [16]. Moreover, other studies have shown that on-site sample preparation extraction systems/kits could be a good option for the diagnostic process in the field [18,28,29,30], offering the advantage of conducting a fast nucleic acid extraction process that helps in the rapid detection of pathogens. Interestingly, direct qPCR amplification of diluted sample materials using the standard CFX 96 PCR system delivered comparable qualitative results for the tested samples. ASFV genomes were detected via direct PCR amplification in all samples with high to moderate viral loads (Ct < 30). Only samples with low viral genome loads (CT > 30) scored mostly negative results (one positive out of seven samples). Nevertheless, 36 samples (out of 42) were detected as positive overall with direct PCR.

Furthermore, the performance of the standard real-time CFX 96 PCR system was compared with three portable qPCR systems (IndiField, Liberty16, and UF-300 Genechecker) based on the isolated DNA samples. For the IndiField cycler, the commercially available lyophilized PCR reagents and the recommended temperature profiles were used. The lyophilized master mix allows the addition of 20 µL of a DNA template, and the kit-based protocol has a total run time of 56 min. This relatively long PCR run time compared to the other POC cyclers, and the 8-fold amount of template, most likely account for the high analytical sensitivity of this workflow among the other thermocyclers. For the other two POCTs on the Liberty16 and the Genechecker, respectively, the ASF p72 gene real-time PCR assay of Haines et al. (2013) was used [11]. These two portable cyclers were tested with a maximum time-reduced temperature profile (37 min for Liberty16 and 19 min for UF-300). For both workflows, the extracted ASFV DNA was amplified and detected with high sensitivity (at least 39 of the 42 samples were positive) despite the short overall PCR run times. Only direct qPCR of the weakly positive samples showed negative results. Surprisingly, the tube-based Liberty16 system with a 37 min runtime performed slightly worse (31 of the 42 samples were positive) than the extremely fast microchip-based UF-300 system with a 19 min runtime (35 of the 42 samples were positive).

The different data suggested that portable molecular assays can be used to detect ASFV DNA in realistic sample materials almost as efficiently as laboratory-based methods. It is clear that a maximum reduction in the PCR run time, as well as the lack of elimination of inhibitory factors when using direct qPCR, must lead to a reduced sensitivity of these methods. However, it is also clear that for diagnostics of clinically affected animals, a maximized sensitivity is in most cases not necessary. Here, according to the diagnostic requirements, a compromise between time and sensitivity must be found. Thus, a negative result in POCT must always be critically reviewed, and testing for freedom of disease in herds or individual animals can be generally limited. Nevertheless, the question of individual testing in the field can be answered positively with the best POC method presented here (Triple*E* + IndiField), as it can provide very comparable results compared to the standard laboratory method. These best POCT results were reached with the portable Triple*E* extraction system combined with the IndiField amplification. A high level of correlation (*R*^2^ > 0.95) was observed between this POCT workflow and the laboratory-based reference method (IndiMag 48 extraction system followed by amplification on Bio-Rad CFX 96) (Figure 3). Moreover, pre-filled reagents of the Triple*E* extraction system could be stored at room temperature for months, and the qPCR reagents of the IndiField assay are lyophilized, which also eliminates the need for a cold chain here.

Some portable molecular assays for rapid on-site detection of African swine fever (ASF) have been described [12,31,32,33,34,35], and a few have been also evaluated in the field [36]. Other studies have demonstrated the importance of applying POCT as a molecular tool in the field, which may even reduce the workload for central laboratories [18,37]. In accordance with our study, the study of Daigle et al. (2020) showed and confirmed the applicability of a portable molecular assay in the field, which was successfully performed with clinical sample materials [18]. Our field molecular assay offers rapid and sensitive DNA/RNA extraction for eight samples in parallel within 10 min, and the possibility to detect ASFV genomes with different POC cyclers and assays. A portable assay may also be carried out in a small mobile laboratory or in a vehicle, avoiding the need to move instruments inside a possibly infected farm.

The limitations of molecular POCT should be presented with caution, and always in the context of the aims of investigations. The expected high diagnostic specificity of POCT can be used to define positive results with certainty. The reduced analytical sensitivity of POCT compared to routine laboratory-based methods may result in some difficulties for the free testing of samples. However, such weakly positive samples are unlikely to be found in diseased pigs, but may be an issue for wild boar carcasses in poor condition.

In summary, the presented data in this study showed that the universal Triple*E* electricity-free extraction system achieved a similar sensitivity to standard automated extraction such as the IndiMag 48, although the obtained Ct values were slightly lower with the standard method. In addition, ASFV genome detection was significantly less sensitive with direct qPCR amplification. This is evident from the analysis of both qualitative and quantitative PCR data. However, it appeared that direct qPCR amplification could be a sufficiently reliable POCT under certain circumstances (e.g., in clinically diseased animals). For molecular POCT, the portable PCR machines tested here using ultra-rapid temperature profiles are generally suitable, and provide comparable results to Bio-Rad’s standard laboratory-based cycler.

## 5. Conclusions

The combination of a portable qPCR system and a manual extraction method resulted in a user-friendly, sensitive, and specific field-deployable diagnostic system. This would help the diagnosis process in remote areas, and could also reduce the amount of field samples that need to be shipped to central laboratories. The application of ASFV-direct qPCR in the field could be an alternative option for samples with high viral genome loads.

## Figures and Tables

**Figure 1 viruses-14-02827-f001:**
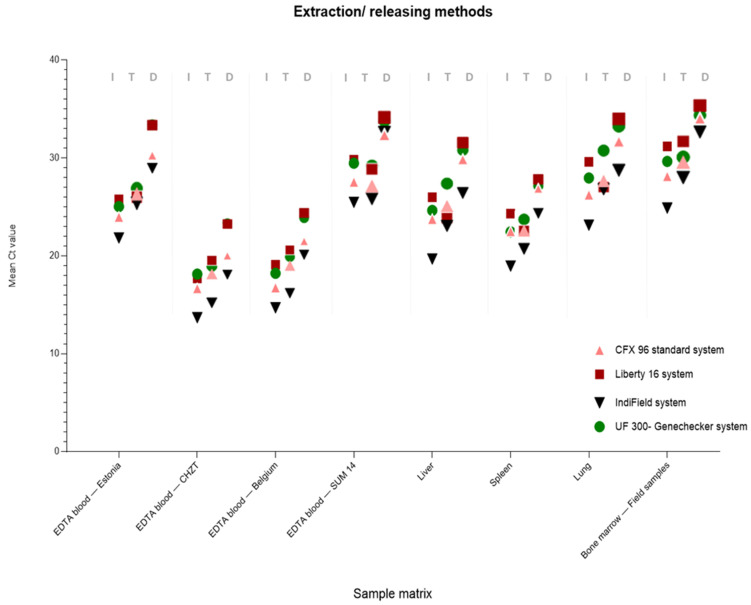
Comparison of the four different qPCR systems based on the three different extraction/releasing procedures. (I = standard automated IndiMag 48 extraction system; T = Triple*E* manual extraction system; D = released DNA amplified via direct qPCR). For the analyses, the qPCR results of different sample matrices from 22 animals, infected with several ASF virus strains, were used (see detailed information in the Methods and Materials section).

**Figure 2 viruses-14-02827-f002:**
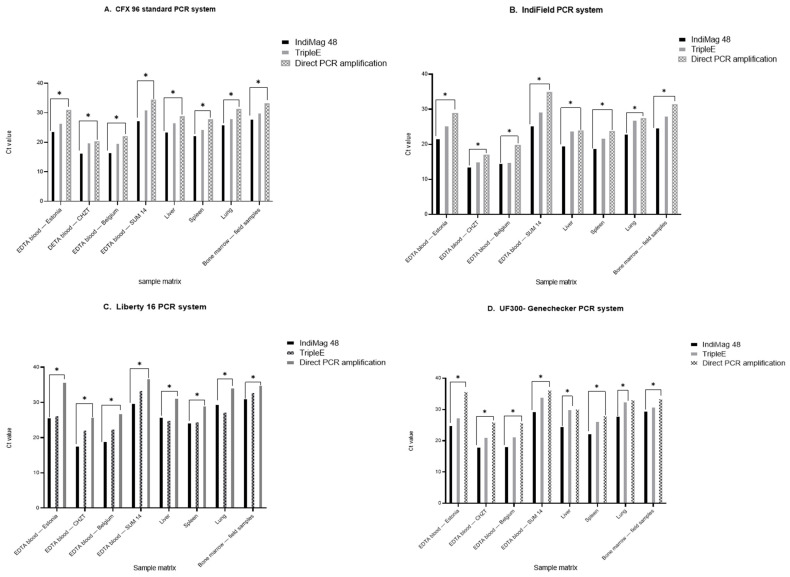
Statistical analyses of the different DNA extraction/releasing methods based on different real-time PCR systems. Based on the different sample matrices, an unpaired *t*-test was performed to test the significance of the different extraction/releasing methods. The IndiMag 48 and Triple*E* system showed highly significant Ct values compared to the direct qPCR amplification (* *p* < 0.01). However, there was no significant difference between the two extraction systems (IndiMag 48 and Triple*E*), and this was presented with a *p*-value > 0.99 (ns). (**A**) Standard deviation (SD) analysis was carried out for all DNA extraction/releasing methods, based on the standard CFX 96 PCR system. The SD value for IndiMag 48 was 4.44, 4.26 for Triple*E*, and 5.06 for direct PCR. (**B**) Based on the IndiField PCR system, the SD value for IndiMag 48 was 4.40, 5.55 for Triple*E*, and 5.92 for direct PCR amplification. (**C**) Based on the Liberty16 PCR system, the SD value for IndiMag 48 was 4.95, 4.31 for Triple*E*, and 4.18 for direct PCR amplification. (**D**) Based on the UF 300-Genechecker system, the SD value for IndiMag 48 was 4.59, 4.83 for Triple*E*, and 4.18 for direct PCR amplification.

**Figure 3 viruses-14-02827-f003:**
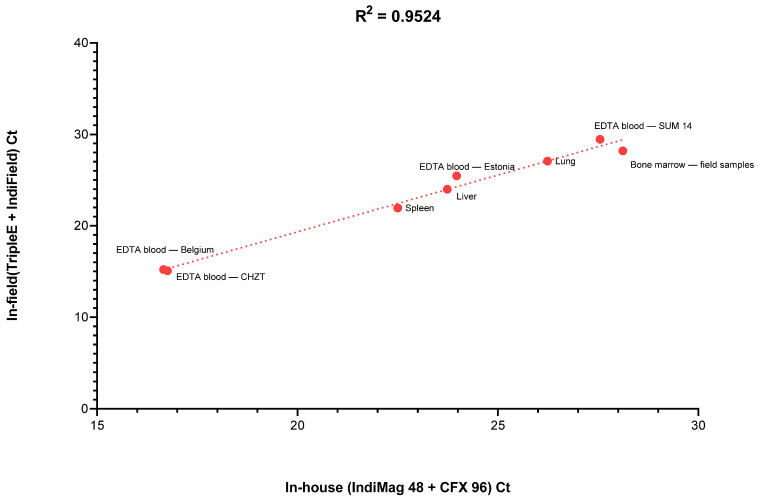
Comparison of the mean Ct values obtained from the same samples tested on the portable in-field system (Triple*E* + IndiField PCR system) versus the standard laboratory-based system (IndiMag 48 + CFX 96 PCR system). A Spearman correlation coefficient test was performed using GraphPad Prism 8 (GraphPad Software Inc., San Diego, CA, USA).

**Table 1 viruses-14-02827-t001:** Qualitative data analysis of the PCR results representing the different extraction methods and qPCR thermocycler. In each column, the number of positive results related to the total number of tested ASFV-positive samples are presented.

	IndiMag 48					Triple*E*					Direct PCR				
	G I	G II	G III	G IV	Total	G I	G II	G III	G IV	Total	G I	G II	G III	G IV	Total
CFX 96	11/11	11/11	13/13	5/7	40/42	11/11	11/11	13/13	5/7	40/42	11/11	11/11	13/13	1/7	36/42
IndiField	11/11	11/11	13/13	7/7	42/42	11/11	11/11	13/13	5/7	40/42	11/11	11/11	13/13	0/7	35/42
Liberty16	11/11	11/11	13/13	6/7	41/42	11/11	11/11	13/13	5/7	40/42	11/11	11/11	9/13	0/7	31/42
UF-300	11/11	11/11	13/13	6/7	41/42	11/11	11/11	13/13	4/7	39/42	11/11	11/11	13/13	0/7	35/42

G I = Group I represents samples with Ct values between 15 and <20; G II = Group II with Ct values between 20 and <25; G III = Group III with Ct values between 25 and <30; G IV = Group IV with Ct values >30 (for detailed raw data, see Appendix A).

**Table 2 viruses-14-02827-t002:** Comparison of extraction/releasing time, qPCR run time, and total processing time for the tested extraction/releasing methods and real-time PCR cyclers (in min).

	IndiMag 48			Triple*E*			Direct PCR		
	Extraction Time	PCR Run Time	Total Processing Time	Extraction Time	PCR Run Time	Total Processing Time	Releasing Time	PCR Run Time	Total Processing Time
CFX 96	31	76	107	10	76	86	5	76	81
IndiField	31	54	85	10	54	64	5	54	59
Liberty16	31	37	68	10	37	47	5	37	42
UF-300	31	19	50	10	19	29	5	19	24

## Data Availability

The dataset used and/or analyzed during the current study is available from the corresponding author upon reasonable request.

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
