# Peer review of "Point-of-Care Testing for Sensitive Detection of the African Swine Fever Virus Genome"

_viruses, 2022, doi:10.3390/v14122827_

Round 1
Reviewer 1 Report
African swine fever is a very problematic disease. With a high degree of transmissibility and a mortality rate that can raise up to 100% of all infected animals, it can cause enormous economic losses. In most developed countries with appropriate resources there are effective surveillance networks and diagnostic laboratories. However, this is often not true for isolated regions or developing countries. Rapid and precise point of care testing is an important and useful tool in these settings, and even in well-resourced countries, it can help in the surveillance activities improving detection and preventing the spreading of outbreaks.
The authors present a rapid and portable ASFV detection system aimed at point of care testing and a direct comparison against a gold-standard real-time qPCR diagnostic technique. The study is well designed and all important details, including methods, results, and conclusions, are well presented in the text. I do not have major concerns. Minor comments and suggestions follow.
Section 2.1. in Materials and Methods: the counting of the number of samples is confusing. In the first line the authors state that 34 samples were used. Then, in lines 110 to 114 the numbers add up to 40 samples. Finally, in the last line the authors state that 42 samples were used. Please check the numbers and correct the text appropriately.
Lines 264 to 266, foot text in the Figure 2 and lines 388 to 390 in the foot text of Figure 3: the text in all these lines disclose the statistical methods used in the study. Since this is a description of the methods, I suggest moving all this to a new section under Materials and Methods describing specifically the statistical methodology used.
Line 407 "... the free testing of samples": I assume that here the authors mean “to test that the animals are free of disease”. Please, rewrite the sentence so it is stated more clearly.
Reviewer 2 Report
The article touches upon an important issue about the molecular diagnosis of ASF in the field. There is no doubt that rapid and specific diagnosis is the key to success in the timely elimination of ASF outbreaks. Unfortunately, the examination of samples in the laboratory often takes a long time. The introduction of field diagnostic methods will contribute to earlier detection of ASF outbreaks. This study is of interest from the point of view of ASF diagnosing, as well as investigating ASF outbreaks at various sites, especially in the field conditions without access to electricity.
The article is written in a good and understandable language. The methods used by the authors correspond to the objectives of the study.
I dont have any significant questions or comments on the presented work. There is an error on the line 44 in the word “Romania”.
Reviewer 3 Report
The African swine fever virus is causing a devastating pandemic affecting the pig industry in many countries of Europe and Asia. There is currently no effective and safty vaccines or therapy for ASF; hence, early and swift detection is critical for any control measures.
The authors have experience in ASFV experiments, in development and validation of diagnostic tools. The article is well written, and the methodology used is adequate. The article is of interest to specialists in laboratory diagnostics and worthy of publication.
However, due to the fact that the article is aimed at еру comparison of the diagnostic tool under special field conditions, only eight field samples may not be enough. It would be interesting to add field samples from outbreaks associated with less virulent ASFV isolates (Estonia 2014).